# ADAPTING TO DISTRIBUTION SHIFT BY VISUAL DOMAIN PROMPT GENERATION

**Zhixiang Chi**[1]*,  **Li Gu**[2]*,  **Tao Zhong**[1],  **Huan Liu**[3],  **Yuanhao Yu**[3],
**Konstantinos N Plataniotis**[1],  **Yang Wang**[2]
[1] University of Toronto, [2] Concordia University, [3] McMaster University
✉ zhixiang.chi@mail.utoronto.ca

## ABSTRACT

In this paper, we aim to adapt a model at test-time using a few unlabeled data to address distribution shifts. To tackle the challenges of extracting domain knowledge from a limited amount of data, it is crucial to utilize correlated information from pre-trained backbones and source domains. Previous studies fail to utilize recent foundation models with strong out-of-distribution generalization. Additionally, domain-centric designs are not flavored in their works. Furthermore, they employ the process of modelling source domains and the process of learning to adapt independently into disjoint training stages. In this work, we propose an approach on top of the pre-computed features of the foundation model. Specifically, we build a knowledge bank to learn the transferable knowledge from source domains. Conditioned on few-shot target data, we introduce a domain prompt generator to condense the knowledge bank into a domain-specific prompt. The domain prompt then directs the visual features towards a particular domain via a guidance module. Moreover, we propose a domain-aware contrastive loss and employ meta-learning to facilitate domain knowledge extraction. Extensive experiments are conducted to validate the domain knowledge extraction. The proposed method outperforms previous work on 5 large-scale benchmarks including WILDS and DomainNet.

## 1 INTRODUCTION

The superior performance of deep models relies on identical distribution between training and testing data (Choi et al., 2018). However, in reality, expecting the training set to cover the universal distribution is practically unfeasible. Consequently, domain shift occurs at test time on unseen distributions, leading to performance deterioration (Koh et al., 2021). To tackle domain shift, (Zhong et al., 2022; Wu et al., 2024) adopts an additional learning phase to adapt the trained model using only a few unlabeled data. Generally, a limited amount of unlabeled data from the target domain conveys *underlying domain distribution* (Zhang et al., 2021a), and is readily accessible at test-time (e.g., image collected during camera calibration). The adaptation is performed only *once for each target domain* instead of every data, making it efficient and practical for deployment on resource-constrained devices. We refer to such a setting as Few-Shot Test-Time Domain Adaptation (FSTT-DA).

In FSTT-DA, extracting domain-specific knowledge from few-shot data remains a major obstacle. Hence, leveraging generalized semantic knowledge from *pre-trained backbones* and transferable domain knowledge from *source domains* is vital. To achieve this, three key challenges must be carefully addressed: 1) how to leverage pre-trained backbones with strong out-of-distribution generalization? 2) how to effectively learn transferable domain knowledge from source domains? 3) how to utilize this knowledge to acquire valid domain-specific knowledge for the unseen target domain?

(Zhong et al., 2022) encodes the multi-source domain knowledge into a mixture of expert networks. A few unlabeled data are collected from a target domain to query the related knowledge to finetune a student network. However, storing domain knowledge in the parameter space of expert networks introduces exorbitant computation and storage costs that grow linearly with the number of source domains (Puigcerver et al., 2023). Moreover, updating the entire network at test-time is infeasible

---

*The authors contributed equally to this work. Project page: https://chi-chi-zx.github.io/VDPG_ICLR24

for resource-constrained devices (Cai et al., 2020; Lin et al., 2022). Alternatively, Visual Prompt Tuning (VPT) (Bahng et al., 2022; Jia et al., 2022; Wang et al., 2022b;a; Han et al., 2023) is adopted to efficiently handle domain shifts. A small number of learnable parameters on the input side, named visual prompts, enable the modification of domain knowledge exclusively. (Zheng et al., 2022) independently encodes the knowledge of each source domain into distinct domain prompts. A target prompt is then produced by linearly combining the pre-learned source domain prompts. (Gan et al., 2023) heuristically partition the source domain prompts into domain-specific and domain-shareable components. During adaptation, a manually crafted regularization term is employed to preserve the domain-shareable part while allowing the domain-specific component updates.

However, several major drawbacks are observed. First, large foundation models (FMs, e.g., CLIP (Radford et al., 2021)) with more powerful out-of-domain generalization is not adopted as the visual backbone. Leveraging rich generalized semantic features from pre-trained FMs has been essential to address domain shifts (Zhang et al., 2021b). However, adapting FMs using only few-shot data in FSTT-DA is challenging. Directly adopting FMs into the above-mentioned methods is not ideal as accessing the FM weights is needed for finetuning or gradient calculation. Recent research has shown that finetuning greatly hampers its robust generalization capability (Wortsman et al., 2022b). On the other hand, it is not applicable when the FMs are only available via APIs in a black-box setting (Oh et al., 2023; Ouali et al., 2023). Second, they learn the source knowledge via cross-entropy loss solely. The domain expert/prompt may inadvertently learn semantic patterns as a shortcut to fulfill the classification task instead of learning transferable domain knowledge (Li et al., 2023). Last, separating the process of modelling source knowledge and the process of learning to adapt into independent training phases may compromise knowledge transfer to unseen domains.

In this work, we aim to adapt FMs to tackle domain shifts in a practical FSTT-DA setting. FMs trained on web-scale data, already exhibit robust generalized features (Wortsman et al., 2022b). We contend that with proper domain-specific guidance, such generalized visual features can boost the model's performance on a specific data distribution. Thus, we propose to build adaptation on top of their features while keeping their inherent robustness unspoiled. Specifically, we propose a prompt generator to generate a domain-specific visual prompt for each target domain to complete such feature guidance. We tackle the above-mentioned knowledge modelling and transfer challenges as follows: we propose a learnable knowledge bank that is shared across source domains to encode their transferable knowledge. Given a target domain, our prompt generator treats a *mini-batch of unlabeled data* as a condition to condense the knowledge bank into a domain-specific prompt. This generated domain prompt is then incorporated into the FM features via a guidance module to process all data in that domain. We train all the modules simultaneously to allow the knowledge bank to automatically explore and learn the transferable source knowledge. Moreover, we employ episodic learning to align the learning objectives of the prompt generator and the guidance module at the meta-level to ensure valid domain-specific guidance (Hospedales et al., 2021). To purify the domain-specific knowledge and reduce semantic interference, we introduce a domain-aware contrastive loss to enhance the prompt generator. This also allows us to further elevate the generator by exploiting large-scale unlabeled data with only domain IDs (Sagawa et al., 2022). Our proposed method does not require access to FM's weights, allowing it to be flexible in a black-box setting (Ouali et al., 2023). It also significantly reduces memory overheads and privacy leaks (Xu et al., 2020b) making it well-suited for on-device deployment scenarios with gradient-free adaptation.

We dub our method as **V**isual **D**omain **P**rompt **G**enerator (VDPG), and summarize our main contributions as follows: 1) We propose a novel approach that employs the visual prompt and the foundation model to tackle distribution shifts. Our approach formulates the adaptation process as generating a visual prompt that encodes domain-specific knowledge to guide the foundational model. 2) we propose domain-aware contrastive loss and meta-learning to facilitate domain knowledge extraction. 3) We conduct extensive experiments to show the superiority of the proposed method among the state-of-the-art and verify the effectiveness of each component.

## 2 RELATED WORK

**Distribution shift.** Domain Generalization (Zhou et al., 2022a) aims to learn a model that can perform well on all unseen target domains by learning a domain-invariant feature representation (Li et al., 2018c;b), exploiting data augmentation strategies (Zhou et al., 2020b;a) or exposing domain

shifts during training via meta-learning (Li et al., 2018a; Balaji et al., 2018). However, deploying the generic model to all unseen target domains fails to explore domain specialty and yields sub-optimal solutions (Sun et al., 2020). Unsupervised Domain Adaptation allows the model to access unlabeled target data at training time (Zhang, 2021). However, the assumption of co-existing source and target data is inapplicable in privacy-sensitive situations where the target domain is inaccessible in advance (An et al., 2022). In contrast, our work exploits few-shot unlabeled target data at test-time to adapt the pre-trained model to unseen target domains (Zhong et al., 2022).

**Test-time adaptation (TTA)**. TTA constructs unsupervised objectives to update the model with unlabeled data before inference. Most previous works focus on entropy minimization (Wang et al., 2021), pseudo-labeling (Liang et al., 2020), auxiliary tasks (Sun et al., 2020; Liu et al., 2023; Chi et al., 2021; Liu et al., 2022b), contrastive learning (Chen et al., 2022c; Wu et al., 2023) and class prototypes (Yang et al., 2020). In addition, TTA is introduced to tackle distribution shifts (Liang et al., 2023) and can be broadly divided into offline and online settings (Gao et al., 2022). In offline TTA, the model uses all target data for multi-epoch adaptation before inference. In online TTA, the model adapts and infers on each target data at the same time. In contrast, considering the constrained resources in the deployment, our work focuses on a more challenging case in which only a small mini-batch of data for each target domain can be used for test-time adaptation (Zhong et al., 2022).

**Foundation models for domain generalization**. The large foundation models (e.g. CLIP ) pre-trained on web-scale data achieve strong out-of-distribution (OOD) generalization(Radford et al., 2021). However, finetuning CLIP's image encoder with task-specific data deteriorates the intrinsic robustness to distribution shifts (Wortsman et al., 2022a). Recent works focus on robust finetuning strategies including two-stage linear probing (Kumar et al., 2022), model ensemble (Wortsman et al., 2022a;b), and contrastive fine-tuning (Goyal et al., 2023; Shu et al., 2023). Instead, to minimize the training cost and to maintain the OOD generalization, our work opts not to finetune the CLIP's image encoder in training and adaptation. Moreover, (Zhou et al., 2020b) employs prompt tuning (Lester et al., 2021) on CLIP, optimizing the prompt vectors prefix to the input tokens of CLIP's text module to enhance performance on downstream tasks. To improve generalization to OOD data, the prompt vectors are conditioned on image inputs at test time (Zhou et al., 2022b; Zhang et al., 2021b; Shu et al., 2022; Derakhshani et al., 2023). However, since these works rely on the prompt prepended to the text input, they incur additional computational costs from the text encoder during inference and are constrained in Vision-Language architectures. Instead, our method generates a visual prompt for CLIP's image encoder, offering flexibility across various vision foundation models.

**Visual prompt**. Visual Prompt Tuning offers a parameter-efficient fine-tuning framework. Learnable prompt tokens are inserted into the visual input, including image pixels (Bahng et al., 2022) and image patch features (Wang et al., 2022b;a; Jia et al., 2022; Huang et al., 2023). Those prompts are trained to adapt pre-trained models to downstream tasks. Furthermore, (Oh et al., 2023; Ouali et al., 2023) introduced a black-box setting that relies solely on pre-computed image and text features, bypassing the need to access the backbone's weights. Our research adopts this practical setting, sidestepping the prohibitive training costs associated with gradient backpropagation through the foundational model. In addition, a concurrent study introduces a prompt generation network that aims to produce input-dependent visual prompt tokens (Loedeman et al., 2023). However, this approach still necessitates access to the backbone's internal architecture and has not been validated in the Few-shot Test-Time Domain Adaptation setting.

## 3   PRELIMINARIES

**Problem setting.** In this work, we follow the Few-Shot Test-time Domain Adaptation (FSTT-DA) as in (Zhong et al., 2022). Specifically, the training set is comprised of $N$ source domains: $\mathcal{D}_s = \{\mathcal{D}_s^n\}_{n=1}^N$, and the test set contains $M$ target domains: $\mathcal{D}_t = \{\mathcal{D}_t^m\}_{m=1}^M$. Each source domain contains labeled data: $\mathcal{D}_s^n = (x_s, y_s)^n$, while the target domains only have unlabeled data: $\mathcal{D}_t^m = (x_t)^m$. $(x, y)$ denotes the input image and output label pair. We assume distribution shift occurs between any pair of the source and target domains $\{\mathcal{D}_s^1, ..., \mathcal{D}_s^N, \mathcal{D}_t^1, ..., \mathcal{D}_t^M\}$, but all the domains share the same label space. FSTT-DA aims to perform training on the source domains. When an unseen target domain $\mathcal{D}_t^m$ is encountered during testing (e.g., model deployed at a new scene), the trained model is expected to adapt to that particular domain using only **a few-shot of unlabeled**

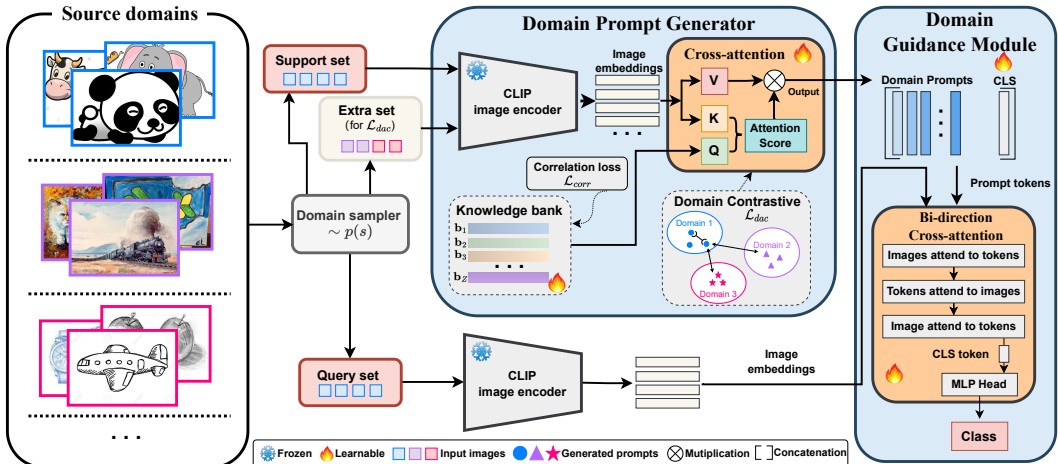

Figure 1: Overview of training pipeline of VDPG. Two disjoint support and query sets are sampled from a training domain. The support set is passed to a domain prompt generator to condense the learned knowledge bank into a domain-specific prompt. The generated prompt is then evaluated on the query set by guiding their feature via a guidance module. Noted, the image/prompt with the same colour belongs to the same domain.

**data**. The adapted model is then adopted to test all data in that domain. FSTT-DA enforces mutual independence between source and target domains, accessing both types of domains is prohibited.

**Motivations.** FMs trained on web-scale datasets with contrastive loss has been observed with strong OOD generalization (Goyal et al., 2023). However, expecting an undifferentiated model to tackle myriad domain shifts in the real world is manifestly less than the ideal (Zhong et al., 2022). Adapting a large model towards unseen distribution is challenging. And the problem becomes tougher when only **a few unlabeled data** is available. Finetuning such a large model under low data availability regimes causes significant overfitting and hampers the intrinsic strong generalization of FMs (Wortsman et al., 2022c). In Table 2, we show that CLIP-based models in general outperform others, and its zero-shot results indicate strong generalization capability even without seeing this particular dataset. It implicitly shows that the visual feature encoded by the CLIP image encoder contains rich generalized semantic features. In contrast, finetuning the CLIP image encoder weights (ERM in Table 2) shows a significant performance drop. Inspired by the aforementioned, our method builds on top of CLIP visual features to improve its performance towards specific unseen domains. It motivates us to propose a method that extracts purified domain-specific knowledge from a few unlabeled data to guide the CLIP visual feature under domain shifts.

## 4 METHOD

**Overview.** In this work, we aim to incorporate the visual prompt and CLIP visual encoder to adapt to unseen target domains. Concretely, we build a learnable knowledge bank to encode transferable knowledge from source data. To adapt to each domain, we propose a domain prompt generator that is conditioned on few-shot unlabeled data to condense the knowledge bank into a domain-specific prompt. A guiding module is then proposed to incorporate the domain-specific knowledge into the CLIP features. Noted, the domain prompt is generated *once* and *reused* for all subsequent inferences on the remaining target data. We employ meta-learning to learn the process of modelling source domains and the process of learning to adapt simultaneously. Fig. 1 demonstrates the overview.

### 4.1 MODEL ARCHITECTURE

**Transferable knowledge bank:** Learning valid source domain knowledge that is suitable for transferring to unseen domains is pivotal. Instead of modelling each of the source domains independently (Zheng et al., 2022; Zhong et al., 2022), we propose a shared space for learning across

all source domains. Concretely, we propose a learnable knowledge bank (KB) $\mathbf{B} \in \mathbb{R}^{Z \times d} = \{\mathbf{b}_1, \mathbf{b}_2, ..., \mathbf{b}_Z\}$ with $\mathbf{b}_z \in \mathbb{R}^d$. $d$ matches the image embedding dimension CLIP (e.g., $d = 1024$ for ViT-L). $Z$ is a source domain-dependent hyper-parameter that controls the KB size. Despite of the shared knowledge, we impose each of $\mathbf{b}_z$ to maintain its own specialty, such as unique visual attributes (Wiles et al., 2021). Therefore, we propose to reduce the correlation between every pair of $\mathbf{b}_*$ by reducing their inner product (enforcing orthogonality) as:

$$\mathcal{L}_{corr}(\mathbf{B}) = \left\| \mathbf{B}\mathbf{B}^T - \text{diag}(\mathbf{B}\mathbf{B}^T) \right\|_2, \tag{1}$$

where diag() keeps only the diagonal entries and $\|\cdot\|_2$ is the L2 loss.

**Conditional domain prompt generator:** Given a domain $\mathcal{D}$, the key contribution of VDPG is to obtain its domain-specific information from a few unlabeled data $\mathbf{x}$. We use boldface $\mathbf{x}$ as a small batch of images and use $x$ to indicate one image. Inspired by conditional generative models (Rombach et al., 2022; Karras et al., 2019), we model such process as conditional distribution $p(\mathcal{D}|\mathbf{x})$. Recall that our method is built upon rich CLIP image embeddings, we denote it as $\mathbf{E}(\mathbf{x}) \in \mathbb{R}^{|\mathbf{x}| \times l \times d}$, where $l$ is the number of embeddings. Our goal is to condense the relevant information from source knowledge $\mathbf{B}$ to facilitate the domain-specific knowledge extraction. To this end, we build a conditional domain prompt generator, $G$, based on cross-attention mechanism (Jaegle et al., 2021; Vaswani et al., 2017) to effectively generate the domain prompts for domain $\mathcal{D}$:

$$\mathbf{P} = AvePool(G(\mathbf{E}(\mathbf{x}), \mathbf{B})) = \text{Attention}(Q, K, V) = \text{softmax}(\frac{QK^T}{\sqrt{h}})V,$$

$$Q = \mathbf{B}W_Q, K = \mathbf{E}(\mathbf{x})W_K, V = \mathbf{E}(\mathbf{x})W_V, \quad W_Q, W_K, W_V \in \mathbb{R}^{d \times h}. \tag{2}$$

$W_Q, W_K, W_V$ are the attention weights and $h$ is their feature dimension. Note, we apply average pooling on the output at the batch dimension to obtain a domain prompt $\mathbf{P} \in \mathbb{R}^{Z \times d}$.

**Domain-aware contrastive loss:** The generated domain knowledge should be category agnostic and exhibit intra-domain similarity and inter-domain discrepancy. In other words, the generated domain prompts for images within the same domain should be similar regardless of their categories. In contrast, images across domains should yield domain prompts with larger differences. To this end, we propose a contrastive learning mechanism to further purify the domain prompt generation. Specifically, we construct a contrastive batch $\mathbf{X}$ with images from different domains: $\mathbf{X} = \{x_i\}$, and $d(x_i)$ denotes the domain ID of $x_i$. Each image $x_i$ has its corresponding domain prompt $\mathbf{P}^i = G(\mathbf{E}(x^i), \mathbf{B})$. To contrast among those domain prompts generated from $\mathbf{X}$, we adopt soft-nearest neighbor loss (Frosst et al., 2019) for multiple positive and negative samples as:

$$\mathcal{L}_{dac}(\mathbf{X}) = -\frac{1}{|\mathbf{X}|} \sum_{i=1}^{|\mathbf{X}|} \log \frac{\sum_{i \neq j, d(x_i)=d(x_j), j=1,...,|\mathbf{X}|} \exp(- \left\| (\mathbf{P}^i - \mathbf{P}^j) \right\|_2 / \tau)}{\sum_{i \neq k, k=1,...,|\mathbf{X}|} \exp\left(- \left\| (\mathbf{P}^i - \mathbf{P}^k) \right\|_2 / \tau\right)}. \tag{3}$$

$\|\cdot\|_2$ is the L2 loss, $\tau$ is set to 0.1 to control the pair-wise importance. Eq. 3 pulls the domain prompts for the images from the same domain closer and pushes away those from different domains.

**Domain guidance module:** Once the domain prompt is generated, we aim to utilize its domain-specific knowledge to guide the CLIP image feature $\mathbf{E}(\mathbf{x})$ towards that particular domain. Note, $\mathbf{P}$ represents domain-specific knowledge, and $\mathbf{E}(\mathbf{x})$ is a generalized feature with rich semantic concepts. Instead of a simple pre-pend operation (Zheng et al., 2022), we design a guidance module ($GM$) to mix those two sets of knowledge. To achieve this, we follow (Kirillov et al., 2023) to stack cross-attention layers with two directions for $GM$. We first append a learnable [CLS] token to $\mathbf{P}$ to form a prompt token and pass it with $\mathbf{E}(\mathbf{x})$ to $GM$. The final prediction $y'$ is obtained by processing the output with an MLP layer as:

$$y' = MLP(\texttt{[CLS]}'), \quad \text{where} \quad \texttt{[CLS]}' = GM([\mathbf{P}, \texttt{[CLS]}], \mathbf{E}(\mathbf{x})). \tag{4}$$

A task-specific loss (e.g., CE loss for classification and MSE for regression) is defined as: $\mathcal{L}_{task}(y', y)$. We combine and weight all the losses by $\lambda$ and $\gamma$ as:

$$\mathcal{L}_{all} = \mathcal{L}_{task} + \lambda \mathcal{L}_{corr} + \gamma \mathcal{L}_{dac}. \tag{5}$$

### 4.2 TRAINING AND INFERENCE

**Domain episodic training on source domains:** Proper training is paramount to elevate the learning of transferable knowledge from source domains and to facilitate the adaptation with few-shot images

simultaneously. First, the domain prompt generated by a small number of images should not only overfit to those few-shot data but be applicable to **all data from that domain**. Second, while each adaptation solely focuses on one particular domain, the mechanism of learning to adapt should be generalized across all domains. Conventional ERM training is inferior because it is not domain-centricfor such a challenging few-show setting. Instead, inspired by the framework of meta-learning for few-shot image classification, we adopt episodic learning to address the above challenges (Chi et al., 2022; Chen et al., 2022b;a). We treat the few-shot adaptation for one particular domain as one episode and simulate large-scale and diverse episodes with source domain datasets.

Algo. 1 demonstrates our training pipeline. Specifically, for each episode, we first sample one domain $\mathcal{D}_s^n \sim p(s)$, and then sample two non-overlapping support set $(\mathbf{x}_{\mathcal{S}})$ and query set $(\mathbf{x}_{\mathcal{Q}}, \mathbf{y}_{\mathcal{Q}})$ (L4-5). In addition to the support set, we sample extra $C$ other domains with a batch of data from each of them to form a contrastive batch $\mathbf{X}$ (L6-9). We then generate the domain prompt using the support set. The generated domain-specific knowledge is evaluated on the disjoint query set (L11-12). Finally, the learnable parameters are updated by the 3 loss terms (L13-14).

---

**Algorithm 1** Training scheme for VDPG

**Require:** $\mathcal{D}_s$: source domains; $\alpha$: learning rate; $p(s)$: domain probability; $G$: prompt generator; $\mathbf{B}$: KB; $GM$: guiding module; $C$: number of contrastive domains
1: **// Learning to generate domain prompt and guide CLIP visual feature**
2: **Randomly initialize**: $G, \mathbf{B}, GM$
3: **while** *not converged* **do**
4:     $\mathcal{D}_s^n \sim p(s)$            ▷ Sample a source domain
5:     $(\mathbf{x}_{\mathcal{S}}), (\mathbf{x}_{\mathcal{Q}}, \mathbf{y}_{\mathcal{Q}}) \sim \mathcal{D}_s^n$    ▷ Sample support and query sets
6:     Construct contrastive batch $\mathbf{X}$:
7:        Sample $C$ unique domains from $\{\mathcal{D}_s \setminus \mathcal{D}_s^n\}$
8:        Sample batches from each $C$ domains: $\{\mathbf{x}^{c1}, \mathbf{x}^{c2}, ..., \mathbf{x}^C\}$
9:        $\mathbf{X} \leftarrow \{\mathbf{x}_{\mathcal{S}}, \mathbf{x}^{c1}, ..., \mathbf{x}^C\}$     ▷ Assign data to $\mathbf{X}$
10:    Compute losses: $\mathcal{L}_{dac}(\mathbf{X})$ and $\mathcal{L}_{corr}(\mathbf{B})$
11:    $\mathbf{P} = G(\mathbf{E}(\mathbf{x}_{\mathcal{S}}), \mathbf{B})$        ▷ Generate domain prompt
12:    $\mathbf{y}'_{\mathcal{Q}} = MLP(GM([\mathbf{P}, \texttt{[CLS]}], \mathbf{E}(\mathbf{x}_{\mathcal{Q}}))$    ▷ Evaluate on query set
13:    $\mathcal{L}_{all} = \mathcal{L}_{task}(\mathbf{y}'_{\mathcal{Q}}, \mathbf{y}_{\mathcal{Q}}) + \lambda\mathcal{L}_{corr} + \gamma\mathcal{L}_{dac}$ ▷ Compute and combine losses
14:    $(G, \mathbf{B}, GM) \leftarrow (G, \mathbf{B}, GM) - \alpha\nabla_{(G,\mathbf{B},GM)}\mathcal{L}_{all}$    ▷ Update via SGD
15: **end while**

---

**Training on unlabeled data:** Instance-level labels (categories) are costly to annotate, but recording the domain label is effortless. WILDS dataset (Sagawa et al., 2022) provides a set of large-scale unlabeled data with domain IDs. Our prompt generator operates on the domain level and the knowledge bank is able to explore more. Thus, both the knowledge bank and the prompt generator can take advantage of such unlabeled data to boost the overall domain knowledge generation. We first pre-train on unlabeled data using Eq. 1 and 3 and then perform training in Algo. 1.

**Inference:** Inference involves forward pass only. For each target domain, a few images are used to generate the domain prompt (Eq. 2), which will be used for testing all images in that domain (Eq. 4).

## 5 EXPERIMENT

### 5.1 EVALUATION DATASETS AND IMPLEMENTATION DETAILS

**Datasets and evaluation:** We follow Meta-DMoE and MABN to evaluate VDPG on challenging real-world WILDS (Koh et al., 2021) benchmarks. WILDS provides practical large-scale realistic distribution shifts with diverse domains and imbalanced data (Chen et al., 2021). We perform experiments on 4 image testbeds that contain both classification and regression tasks: iWildCam (Beery et al., 2021), Camelyon17 (Bandi et al., 2018), FMoW (Christie et al., 2018), and PovertyMap (Yeh et al., 2020). We follow official splits in source and target domains, and official metrics: accuracy, Macro F1, worse-case (WC) accuracy, Pearson correlation (r), and its worst-case. We also evaluate DomianNet (Peng et al., 2019) which contains 6 domains with 569K images of 345 classes. We follow the official leave-one-domain-out to train 6 models and report the accuracy. In Appendix c, we show the details of testbeds and compare the imbalance conditions across different benchmarks.

**Architecture:** We adopt the image encoder from CLIP to extract the image embeddings. We use ViT-B/16 for DomainNet and ViT-L/14 for WILDS. Their embedding dimensions ($d$) are 768 and 1024. We stack two cross-attention layers for domain prompt generator $G$, and the feature dimension $h$ for both $G$ and $GM$ is the same as embedding size $d$. The number of attention heads is set to 8.

**Training details:** We perform training using SGD with a batch size of 64 for 30 epochs. The initial learning rates are set to $3e^{-3}$ and $5e^{-4}$ with cosine decay for WILDS and DomainNet. The loss

Table 1: Evaluation on WILDS image testbeds under out-of-distribution setting. Our proposed method performs well on both classification and regression tasks and achieves the best results on 3 out of 4 datasets. ($*$ : results obtained by running their official code without ensembles. $\dagger$ : PovertyMap has input of 8 channels, we use only first 3, leading to only 38% data usage)

| Method | Backbone | iWildCam | | Camelyon17 | FMoW | | PovertyMap (Regression) | |
|---|---|---|---|---|---|---|---|---|
| | | Acc | Macro F1 | Acc | WC Acc | Avg Acc | WC Pearson r | Pearson r |
| ERM | | 71.6 (2.5) | 31.0 (1.3) | 70.3 (6.4) | 32.3 (1.25) | 53.0 (0.55) | 0.45 (0.06) | 0.78 (0.04) |
| CORAL | | 73.3 (4.3) | 32.8 (0.1) | 59.5 (7.7) | 31.7 (1.24) | 50.5 (0.36) | 0.44 (0.06) | 0.78 (0.05) |
| IRM | | 59.8 (3.7) | 15.1 (4.9) | 64.2 (8.1) | 30.0 (1.37) | 50.8 (0.13) | 0.43 (0.07) | 0.77 (0.05) |
| ARM-CML | CNNs | 70.5 (0.6) | 28.6 (0.1) | 84.2 (1.4) | 27.2 (0.38) | 45.7 (0.28) | 0.37 (0.08) | 0.75 (0.04) |
| ARM-BN | | 70.3 (2.4) | 23.7 (2.7) | 87.2 (0.9) | 24.6 (0.04) | 42.0 (0.21) | 0.49 (0.21) | **0.84 (0.05)** |
| Meta-DMoE | | 77.2 (0.3) | 34.0 (0.6) | 91.4 (1.5) | 35.4 (0.58) | 52.5 (0.18) | 0.51 (0.04) | 0.80 (0.03) |
| MABN | | 78.4(0.6) | 38.3(1.2) | 92.4(1.9) | 36.6(0.41) | 53.2(0.52) | **0.56(0.05)** | **0.84(0.04)** |
| Zero-shot (ZS) | ViT-L/14 | 28.7 | 1.0 | - | 13.3 | 21.1 | - | - |
| FLYP$^*$ | CLIP | 72.2 (0.4) | 41.9 (0.3) | - | 46.0 (0.3) | **63.3 (0.4)** | - | - |
| VDPG (Ours) | | **78.8 (0.2)** | **46.5 (0.3)** | **96.0 (0.4)** | 46.4 (0.5) | 61.9 (0.4) | 0.51 (0.03)$^\dagger$ | 0.83 (0.04)$^\dagger$ |

Table 2: Evaluation on DomainNet. Our method significantly outperforms SOTA methods.

| Method | Backbone | Clip | Info | Paint | Quick | Real | Sketch | Avg. |
|---|---|---|---|---|---|---|---|---|
| ERM | | 58.1 (0.3) | 18.8 (0.3) | 46.7 (0.3) | 12.2 (0.4) | 59.6 (0.1) | 49.8 (0.4) | 40.9 |
| Mixup (Xu et al., 2020a) | | 55.7 (0.3) | 18.5 (0.5) | 44.3 (0.5) | 12.5 (0.4) | 55.8 (0.3) | 48.2 (0.5) | 39.2 |
| CORAL (Sun & Saenko, 2016) | | 59.2 (0.1) | 19.7 (0.2) | 46.6 (0.3) | 13.4 (0.4) | 59.8 (0.2) | 50.1 (0.6) | 41.5 |
| MTL (Blanchard et al., 2011) | CNNs | 57.9 (0.5) | 18.5 (0.4) | 46.0 (0.1) | 12.5 (0.1) | 59.5 (0.3) | 49.2 (0.1) | 40.6 |
| SegNet (Nam et al., 2019) | | 57.7 (0.3) | 19.0 (0.2) | 45.3 (0.3) | 12.7 (0.5) | 58.1 (0.5) | 48.8 (0.2) | 40.3 |
| ARM (Zhang et al., 2021a) | | 49.7 (0.3) | 16.3 (0.5) | 40.9 (1.1) | 9.4 (0.1) | 53.4 (0.4) | 43.5 (0.4) | 35.5 |
| Meta-DMoE (Zhong et al., 2022) | | 63.5 (0.2) | 21.4 (0.3) | 51.3 (0.4) | 14.3 (0.3) | 62.3 (1.0) | 52.4 (0.2) | 44.2 |
| MABN (Wu et al., 2024) | | 64.2 | 23.6 | 51.5 | 15.2 | 64.6 | 54.1 | 45.5 |
| DoPrompt (Zheng et al., 2022) | ViT-B/16 IMN | 67.6 (0.2) | 24.6 (0.1) | 54.9 (0.1) | 17.5 (0.2) | 69.6 (0.3) | 55.2 (0.5) | 48.3 |
| Zero-shot (ZS) | | 69.9 | 48.2 | 65.4 | 14.5 | **82.3** | 62.5 | 57.1 |
| ERM (Cha et al., 2022) | ViT-B/16 | 68.0 (0.2) | 22.5 (0.6) | 46.5 (4.2) | 18.5 (0.9) | 58.7 (2.7) | 52.5 (1.2) | 44.4 |
| MIRO (Cha et al., 2022) | CLIP | 74.9 (0.2) | 37.1 (0.4) | 59.8 (0.6) | **18.7 (1.2)** | 72.2 (0.2) | 61.2 (0.9) | 54.0 |
| VDPG (Ours) | | **76.3 (0.2)** | **49.3 (0.1)** | **67.8 (0.1)** | 17.4 (0.2) | 81.5 (0.3) | **66.6 (0.2)** | **59.8** |

weights $\gamma$ and $\lambda$ are set to 0.1. When unlabeled data is used in WILDS, we first train $G$ and **B** and finetune with $GM$. Appendix D provides details of more hyperparameters.

## 5.2 MAIN RESULTS

**Comparison on WILDS:** We compare with CNN-based methods: ERM, CORAL (Sun & Saenko, 2016), IRM (Arjovsky et al., 2019), ARM (Zhang et al., 2021a), Meta-DMoE and MABN (Wu et al., 2024); ViT-based methods: zero-shot CLIP (ZS) (Radford et al., 2021) and FLYP (Goyal et al., 2023). We adopt the results from Meta-DMoE and implement ZS (prompt="category_name" ) and FLYP using their official code, we use the single model of FLYP for a fair comparison. Note, that ZS can not be computed on Camelyon17 as the categories have no semantic meaning and PovertyMap as it is a regression task. As reported in Table 1, our method outperforms SOTA on iWildCam, Camelyon17, and FMoW. It is noted that ZS directly collapses due to the lack of learning on source domain datasets. Additionally, although freezing the backbone, our approach still outperforms the best finetuning-based method, FLYP, by 4.6% on the F1 score of iWildCam. FLYP follows the language-vision training as CLIP, therefore, it is not able to perform the regression task. In contrast, our VDPG is flexible for different learning tasks including regression on PovertyMap. On the other hand, as the CLIP image encoder only accepts 3-channel input, our method performs comparably with ARM and Meta-DMoE while using 3 out 8 channels on PovertyMap with only 38% data utility.

**Comparison on DomainNet:** Table 2 reports the evaluation on DomainNet with a wild range of SOTA methods. We adopt the results from Meta-DMoE and MIRO and implement ZS. In general, the CLIP-based methods perform better than their CNN-based counterparts. However, when finetuning the CLIP backbone (ERM), its performance is even worse than ZS. Instead, our method freezes the CLIP backbone to maintain its OOD generalization. In addition to our strong domain knowledge generator, significant improvement is observed.

**Robustness of in-distribution and out-of-distribution settings:** Table 3 reports the comparison with FLYP for both single and ensembled models under in-distribution (ID) and OOD settings on iWildCam. Although ensembling improves FLYP by a large margin, it is still sub-optimal due to

Table 3: Evaluation on ID and OOD setting on iWild-Cam.

| Methods | ID Acc | ID F1 | OOD Acc | OOD F1 |
|---|---|---|---|---|
| FLYP w/o ensemble | 75.2 | 57.5 | 72.2 | 41.9 |
| FLYP w/ ensemble | 76.2 | 59.9 | 76.2 | 46.0 |
| VDPG (Ours) | **77.2** | **60.2** | **78.8** | **46.5** |

Table 4: Evaluation on the domain knowledge by replacing the domain prompt.

| Domain Prompt | ID F1 | OOD F1 |
|---|---|---|
| Random | 31.3 | 24.5 |
| KB (**B**) | 33.3 | 27.9 |
| Zeros | 45.4 | 37.1 |
| VDPG (Ours) | **60.2** | **46.5** |

the lack of domain-specific knowledge from the target domains. In contrast, our method is able to achieve superior results by incorporating domain-specific knowledge into the CLIP features.

## 5.3 DOES THE GENERATOR REALLY OBTAIN THE DOMAIN-SPECIFIC INFORMATION?

The key contribution of VDPG is to generate high-quality domain-specific prompts that are tailored to each target domain. We validate such property on iWildCam as it has 48 diverse target domains.

**Replacing the generated prompt with others contents:** To show that the generated domain prompts are meaningful, we replace them with different contents, including randomly initialized prompts, KB **B** and zeros. As reported in Table 4, there are significant performance drops for all of them (Fig. 8 in Appendix B.9 shows per-domain comparison). It indicates that our generated domain prompts play an important role in guiding the CLIP image feature. Fig. 2a and 2b shows the t-SNE (Van der Maaten & Hinton, 2008) visualization on the features of CLIP and $GM$. Each point and color represents a sample and a class. It is noted that with our generated domain prompt, the features are better clustered and more discriminative than CLIP features. It indicates that the generated domain prompts are able to guide the CLIP image feature for better performance.

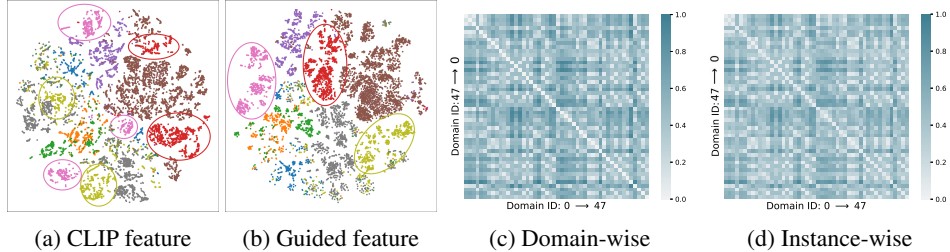

(a) CLIP feature     (b) Guided feature     (c) Domain-wise     (d) Instance-wise

Figure 2: a-b) t-SNE feature visualization of before and after guidance. c-d) Comparison among generated domain prompts on 48 target domains in iWildCam.

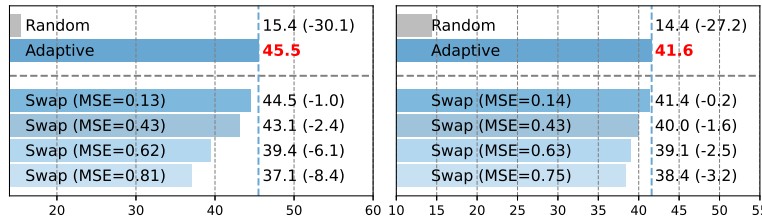

(a) Domain prompt swapping for #40.     (b) Domain prompt swapping for #47.

Figure 3: Swapping the generated domain prompts with various similarity for domain #40 and #47.

**Inter-domain discrepancy and intra-domain similarity:** To show such properties, we compute the L2 distance between every pair of the 48 generated domain prompts, as illustrated in Fig. 2c. It validates that the generated domain prompts are diverse since each domain has its own specialty. We further compare the domain prompt generated among data instances. The L2 distances are averages within each domain. As shown in Fig. 2d, it follows the same pattern as Fig. 2c, further reflecting that the extracted domain-specific knowledge is category-agnostic and only relates to the domain.

Table 5: Ablation on different modules.

| Row | Modules | Training | ID F1 | OOD F1 |
|---|---|---|---|---|
| 1 | ZS | ERM | 1.3 | 1.0 |
| 2 | Linear probing | ERM | 47.2 | 38.7 |
| 3 | Fine-tuning | ERM | 48.3 | 36.4 |
| 4 | Frozen CLIP + $GM$ | ERM | 49.7 | 39.6 |
| 5 | + Generator $G$ | ERM | 46.8 | 32.5 |
| 6 | + KB (full model) | ERM | 47.0 | 32.4 |
| 7 | Full model | Episodic | 54.5 | 43.6 |

Table 6: Ablation on loss and unlabeled data.

| $\mathcal{L}_{corr}$ | $\mathcal{L}_{dac}$ | Unlabeled data | ID F1 | OOD F1 |
|---|---|---|---|---|
| - | - | - | 54.5 | 43.6 |
| ✓ | - | - | 56.7 | 44.1 |
| - | ✓ | - | 58.6 | 44.6 |
| ✓ | ✓ | - | 58.9 | 45.5 |
| ✓ | ✓ | ✓ | 60.2 | 46.5 |

**Correlation among domains:** It is natural to expect correlated domains. Intuitively, less correlated domains should have diverse domain information and vice versa. Thus, if we swap the domain prompts with a larger distance in the prompt, the performance should drop more. Thus, we randomly select two domains and conduct such domain prompt swapping. As shown in Fig. 3, more performance drop is observed when the swapped domain prompts have larger diversity. It demonstrates that our generated prompt can reflect the domain diversity.

## 5.4 ABLATION STUDY

**Ablation on different modules:** We conduct ablation on iWildCam to evaluate each proposed component. As reported in Table 5, linear probing (row 2) can achieve comparable performance, indicating that the visual features from CLIP are quite robust. However, when updating the CLIP's weights, its generalization is decreased (row 3), showing that fintuning could be harmful. Directly processing CLIP features with more parameters only improves slightly (row 4), as the domain-specific knowledge is still missing. Surprisingly, adding the generator and the knowledge bank even deteriorates the performance (row 5&6). We hypothesize that ERM training is not able to provide such supervision at the domain level to learn transferable knowledge and the mechanism of adaptation. Therefore, when episodic learning is employed to enforce domain-centric learning(row 7), the overall performance is boosted.

**Ablation on loss functions and training on unlabeled data:** $\mathcal{L}_{corr}$ enforces each of the vectors in the knowledge bank to maintain its own special feature. And $\mathcal{L}_{dac}$ ensures the discriminative information generated at the domain level. Therefore, optimizing both during training improves the model performance, as demonstrated in Table 6. Furthermore, training the generator and knowledge bank on unlabeled data brings positive improvement, further pushing the performance boundary.

**Computational cost:** Table 7 reports the model size and total computational cost. Note, all methods use the ViT/B-16 model. At train-time, both FYLP and DoPrompt rely on fine-tuning the backbone. At test-time, FYLP exploits the text

Table 7: Comparison on the computational cost on 512 images

| Method | Model size | Finetune | Inference (512 images) |
|---|---|---|---|
| FYLP | 149M | ✓ | 26.2T FLOPS |
| DoPrompt | 88.52M | ✓ | 36.7T FLOPS |
| VDPG | 102M | ✗ | 19.7T FLOPS |

encoder to generate the weights for the classifier head. DoPrompt needs to adapt before making each prediction. In contrast, our method only adapts once with few-shot data (16 images) before making inferences on all target data. We discuss reproducibility and limitation in Appendix A.

## 6 CONCLUSION

In this work, we present VDPG, an approach for adaptation to distribution shift using few-shot unlabeled examples at test-time. We formulate the adaptation process as generating a visual prompt that encodes domain-specific knowledge to guide the foundational model. We introduce a domain-aware model architecture to encode the transferable domain knowledge and devise a meta-learning algorithm to facilitate the domain knowledge extraction. We demonstrate that VDPG outperforms previous works on 5 large-scale benchmarks in DomainNet and WILDS. And our method is efficient in the computation cost at test-time.

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

APPENDIX

## A  REPRODUCIBILITY, LIMITATIONS AND DISCUSSION

**Reproducibility:**  We keep all the implementations and dataset utilization consistent compared with the prior works. We follow the official setting to process the data and use the official splits on source and target domains. The foundation model we use, CLIP, is publicly available, we use the official code to load the model. Our algorithm is straightforward, and all the hyper-parameters used are listed in the implementation section in Section 5.1 and Appendix D. Our source code will be available upon paper acceptance.

**Limitations and discussion:**  One limitation of our method is that the performance highly depends on the out-of-distribution generalization in the foundation models. Although pre-trained on the web-scale dataset, it is not guaranteed that the foundation model covers all the data, especially those that are strictly protected due to privacy. However, our method is built on top of the foundation models without knowing their architectures. It is flexible as long as the final feature can be obtained. Thus, integrating our method with a stronger foundation model is also able to improve our method overall.

## B  ADDITIONAL EXPERIMENTS

### B.1  DIFFERENT BACKBONES ON DOMAINNET

In this section, we conduct the experiment on DomianNet with two different backbones: ViT-L/14 pre-trained with CLIP and ViT-B/14 pre-trained with DINOv2 (Oquab et al., 2023). As shouwn in Table 8, our VDPG outperforms the baselines. Please note that for DINOv2, we only compare our results with two baselines: linear probing and ERM.

Table 8: Evaluation on DomainNet with different foundation model backbones.

| Method | Backbone | Clip | Info | Paint | Quick | Real | Sketch | Avg. |
|---|---|---|---|---|---|---|---|---|
| Zero-shot (a photo of label) | ViT-L/14 CLIP | 79.3 | 53.2 | 72.1 | 22.1 | **87.1** | 72.8 | 64.4 |
| Zero-shot (label) | | 78.1 | 54.0 | 71.6 | 21.8 | 86.0 | 71.2 | 63.8 |
| VDPG (Ours) | | **82.4** | **54.9** | **73.1** | **22.7** | 85.0 | **73.2** | **65.2** |
| Finetuning (ERM) | DINOv2 | 75.4 | 35.7 | 63.2 | 15.8 | 77.6 | 62.0 | 55.0 |
| Linear probing | ViT-B/14 | 74.4 | 37.1 | 66.5 | 13.4 | 78.1 | 65.9 | 55.9 |
| VDPG (Ours) | (Oquab et al., 2023) | **77.2** | **39.6** | **68.1** | **17.6** | **80.4** | **67.7** | **58.4** |

### B.2  IS MODEL ENSEMBLE HELPING?

Ensemble the finetuned CLIP model and the original pre-trained CLIP model can better balance the tradeoff between ID and OOD performance. We also conduct such experiment by simply averaging the 3 models as in  (Goyal et al., 2023). As reported in Table 9, ensembling has less effect compared with FLYP in Table 3.  Please note, that FLYP finetunes the whole image encoder in the source domain and linearly combined with the original CLIP image encoder in the parameter space. The intuition is that the original CLIP has strong OOD capability and the finetuned version contains the data-specific knowledge learned from source domains. The ensemble can be viewed as a fusion of those two sets of knowledge,

Table 9: Evaluation on ensemble strategy.

| iWildCam metric | ID Acc | ID F1 | OOD Acc | OOD F1 |
|---|---|---|---|---|
| Model #1 | 77.4 | 60.3 | 79.0 | 46.8 |
| Model #2 | 76.8 | 59.5 | 78.5 | 46.6 |
| Model #3 | 75.7 | 54.6 | 77.8 | 44.4 |
| Ensembled | 76.7 | 59.7 | 78.6 | 46.9 |

### B.3 DIFFERENT ARCHITECTURE FOR THE GUIDANCE MODULE

We evaluate the impact from different Guidance module by replacing the guidance module with regular transformer layers and performing the pre-pending operation without conditions. As reported from Table 10, conditional operation plays a paramount role in the domain guiding process.

Table 10: Evaluation on different architecture of the Guidance Module.

|  | F1 score on iWildCam |
| --- | --- |
| Bidirectional condition layers | 46.4 |
| Regular self-attention layers | 39.7 |

### B.4 NUMBER OF SOURCE DOMAINS

To show the effect from different numbers of source domains, we conduct such an experiment by treating Clipart as the target domain and randomly selecting the source domains as shown in Table 11. When there is only one source domain, we are not able to compute domain contrastive loss, therefore, the capability to generate domain-specific knowledge is hampered (Liu et al., 2022a). When there are two domains, the generator is enforced to compute domain-specific knowledge, therefore the accuracy is boosted. When more and more source domains are involved during training, the generalization is improved but the overall gain becomes smaller.

Table 11: Evaluation on the number of source domains.

| Number of source domains | 1 | 2 | 3 | 4 | 5 |
| --- | --- | --- | --- | --- | --- |
| Accuracy on Clipart | 55.8 | 71.8 | 74.5 | 75.7 | 76.3 |

### B.5 INSTANCE-WISE COMPARISON ON GENERATED DOMAIN PROMPTS

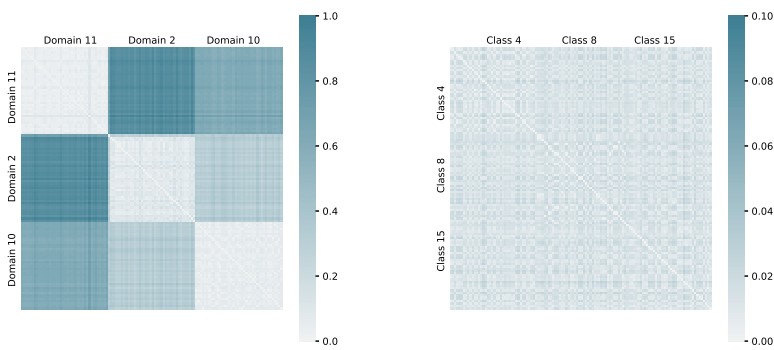

(a) Class 24 samples from different domains (b) Domain 8 samples from different classes

Figure 4: **Comparison among different pairs of instances.** L2 distance between generated prompts from a) samples of class 24 but from different domains; b) Samples all from domain 8 but with different classes.

In Fig. 2d, we present the distance for every pair of data instances but averaged for each domain. In Fig. 4, we randomly select sample instances and show their differences in generated prompts.

In Fig. 4a, we randomly select samples with class label 24, but from different domains. As we can see, even with the same class category, the generated prompts from different domains have larger differences compared to the pairs from the same domain.

On the other hand, in Fig. 4b, we randomly sample data samples from the same domain with different classes. It is evident that, even with the different classes, their generated domain samples are close to each other.

Fig. 4a and Fig. 4b indicate that our domain prompt generator extracts high-quality domain-specific information which is class-agnostic and only related to domains.

### B.6 CORRELATION ON KNOWLEDGE BANK VECTORS

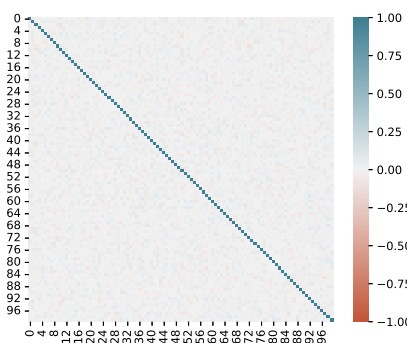

Figure 5: **Correlation among each pairs of $\mathbf{b}_z$ in B. B** is adopted after training on iWildCam dataset with $Z = 100$.

In section 4.1, we aim to allow our network to automatically explore the transferable knowledge while learning how to adapt. We enforce each of the vector $\mathbf{b}_z$ to learn special characters by minimizing $\mathcal{L}_{corr}$ in Eq. 1. Fig. 5 shows such correlation after training. It is observed that the correlation between different pairs of $\mathbf{b}_z$ is small, indicating each of them exhibits their own properties.

### B.7 TRAINING ON UNLABELED DATA:

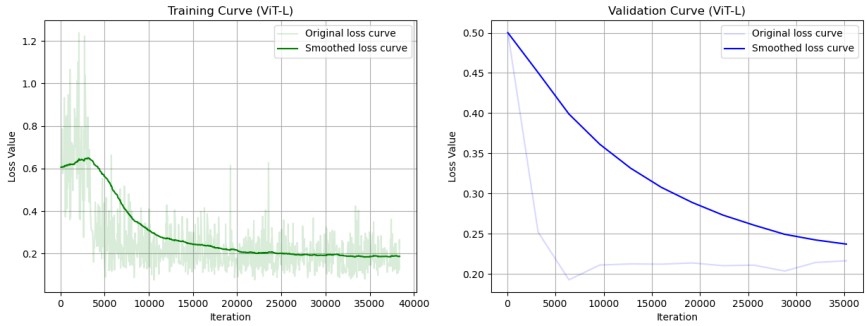

Figure 6: **Training curve when pre-training the generator and knowledge bank on the unlabeled data.**

Fig. 6 shows the training curve on unlabeled data of iWildCam. We use the unlabeled set for training and use the source domains for validation. As we can see, the domain generator is able to generate unseen domains to extract diverse domain-specific knowledge among domains.

### B.8 ABLATION STUDY DETAILS

**Details:** We describe more experimental details of the ablation study in Table 5 as:

- Row 4, Frozen CLIP + GM: since the generated domain prompt is missing in this experiment, we treat only the learnable [CLS] token as the prompt token. The guiding module is kept the same.

- Row 5, + Generator G: the generator originally takes two inputs. Since the knowledge bank is missing in this experiment, we treat the cross-attention as self-attention. All the operation is performed on the image features.

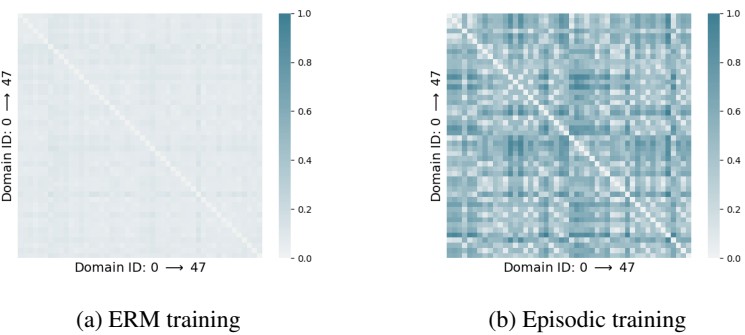

(a) ERM training         (b) Episodic training

Figure 7: **Comparison among different pairs of generated domain prompts from 48 OOD domains of iWildCam.** a) results from ERM training, b) results from Episodic training.

**Comparison on domain prompts for different training schemes:** In the ablation study, we have shown that when adding the domain prompt generator and the knowledge bank, the performance is harmed under ERM training but boosted when episodic learning is utilized. This is because the ERM training is not domain-centric to learn domain-specific information. Fig. 7a and 7b show that episodic training allows our domain prompt generator to extract domain-specific information for better feature guidance to unseen distributions.

### B.9 PER DOMAIN F1 SCORE:

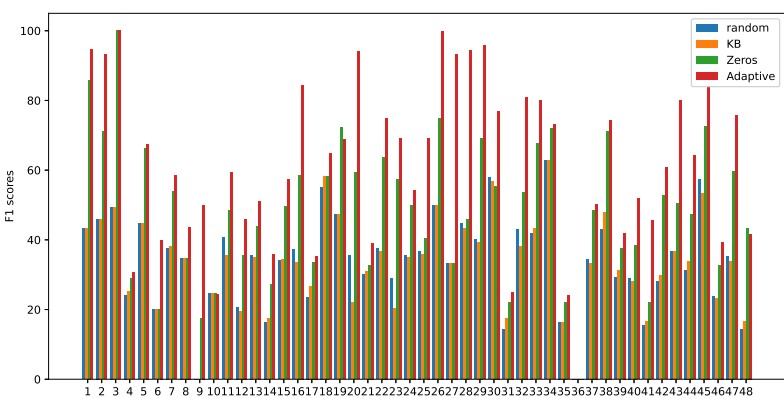

Figure 8: Per domain F1 score for replacing the domain prompt with other contents

## C  DETAILS ON DATASETS

Table 12: Detailed description of 4 image testbeds in WILDS benchmarks.

|  | IWildCam | FMoW | Camelyon17 | PovertyMap |
|---|---|---|---|---|
| Domain meaning | Camera traps | years | hospital | countries |
| # of source domains | 243 | 11 | 3 | 14 |
| # source images | 129,809 | 76,863 | 302,436 | 10,000 |
| # of target domains | 48 | 4 | 1 | 5 |
| # of target images | 42,791 | 22,108 | 85,054 | 4,000 |
| # of categories | 182 | 62 | 2 | regression |

**WILDS benchmarks:** Table 12 reports the details of 4 image testbeds in WILDS benchmarks. Noted, we only list the source (training) and target (testing) domains and images. The validation sets are not included. It shows that those datasets are large-scale at both domain and class level.

Table 13: Comparison on the overview of the popular benchmarks. Imbalance ratio is calculated by max sample numbers/min number of samples within each .

| Dataset | Year | Images | Category | Domain | Imbalance ratio | Data content |
|---|---|---|---|---|---|---|
| Office-Caltech10 | 2021 | 2.5K | 10 | 4 | - | office |
| PACS | 2017 | 9.9K | 7 | 4 | 3.59 | animal and stuff |
| DomainNet | 2019 | 569K | 345 | 6 | 2.35 | Common objects |
| WILDS-iWildCam | 2020 | 203K | 182 | 323 | 3490 | wild animals |
| WILDS-Camelyon17 | 2018 | 450K | 2 | 50 | 38.57 | medical tissue |
| WILDS-FMoW | 2018 | 141K | 62 | 16 | 2891 | satellite |
| WILDS-PovertyMap | 2020 | 19k | regression | 23 | 15.05 | satellite |

**Comparison across benchmarks:** Table 13 compares the existing benchmarks that are popular for domain generalization. We evaluate our method on two largest datasets, DomainNet and WILDS. We also compute the imbalance ratio to demonstrate that WILDS is more challenging. In fact, it is common for some domains to suffer data scarcity. For example, general hospitals accept more patients while specialized hospitals may accept fewer patients, collecting fewer data points.

## D  ADDITIONAL HYPERPARAMETER

Table 14: Hyper parameters for sampling in Algo. 1.

|  | IWildCam | FMoW | Camelyon17 | PovertyMap | DomainNet |
|---|---|---|---|---|---|
| Batch size |  |  | 64 |  |  |
| Support size |  |  | 16 |  |  |
| Query size |  |  | 48 |  |  |
| Contra. domains | 2 | 2 | 1 | 2 | 2 |
| # images / contra. domains | 8 | 8 | 16 | 8 | 8 |
| **X** size |  |  | 80 |  |  |
| # Adaptation images |  |  | 16 |  |  |

**Sampling at training:** Table 14 shows the hyperparameters we used for performing the sampling in Algo. 1. Those hyperparameters are searched on OOD validation of the IWildCam dataset. We simply apply the same to other datasets.

Table 15: Size of knowledge bank for each of the evaluated benchmarks.

| Dataset | Source domain # | Size of knowledge bank B (Z) |
|---|---|---|
| DomainNet | 5 | 5 |
| WILDS-iWildCam | 243 | 100 |
| WILDS-Camelyon17 | 3 | 100 |
| WILDS-FMoW | 11 | 5 |
| WILDS-PovertyMap | 14 | 10 |

**Size of knowledge bank B:** Knowledge bank aims to condense the transferable knowledge from the source domains into the target domain prompt. Intuitively, each source domain should maintain its own specialty. $Z$, which is the size of **B**, should be the same as the number of source domains. However, there could be some correlation among domains, therefore, increasing $Z$ could allow better exploitation from the training dataset, but there is a risk that redundant information will be learned. We set the search space of $Z$ as $\{5, 10, 100, 150, 200\}$, and the value is picked based on validation performance.

