# OpenReview forum: "Adapting to Distribution Shift by Visual Domain Prompt Generation"
_ICLR.cc/2024/Conference — ICLR 2024 poster_

### Official Review · Reviewer_PHNg · 2023-10-26

**Soundness:** 3 good
**Presentation:** 1 poor
**Contribution:** 3 good
**Rating:** 5
**Confidence:** 4

**Summary:**

This paper proposes an approach using pre-trained features of the foundation model. They build a knowledge bank to learn the transferable knowledge from source domains, and then the knowledge bank is used to generate a domain-specific prompt by a domain prompt generator. This prompt as the guidance can enable domain-aware learning.

**Strengths:**

${\bf Strengths:}$

[$\textbf{Well-illustrated Figures}$] The figures can clearly convey the idea of the proposed method.

[$\textbf{Comprehensive Experiments}$] This paper offers the experimental results comparing many baseline methods, ablation study of various losses and hyper-parameter analysis.

**Weaknesses:**

${\bf Weaknesses:}$

[$\textbf{Confusing Expression}$] The expression is sometimes hard to follow and understand. For example, (i) I am not sure how the method works from “The domain prompt then directs the visual features towards a particular domain via a guidance module.”; (ii) they first mention “propose an approach on top of the pre-computed features of the foundation model.”, but then say, “we build a knowledge bank to learn the transferable knowledge from source domains.”. It cannot see the connection between these two sentences. (iii) For “In FSTT-DA, extracting domain-specific knowledge from few-shot data remains a major obstacle.
Hence, leveraging generalized semantic knowledge from pre-trained backbone and transferable domain knowledge from source domain datasets is vital.”, these two sentences have no logic relation. Considering the problems above, it needs a careful refinement before publication.

[$\textbf{The Selection of CLIP Image Encoder}$] The paper uses two different image encoders for DomainNet and WILDS. Could the authors explain the reason? It would be nice to see the performance of two encoders on two datasets.

[$\textbf{Missed Experiments}$] (i) One SOTA work is not compared, e.g., “CLIPood: Generalizing CLIP to Out-of-Distributions, ICML 2023”. (ii) It would be nice to see the computational cost of MIRO.

[$\textbf{Some Suggestions}$] The subfigures in Figure 2 do not show the obvious differences between baseline and the proposed method. It would be nice to change style or type of figures to make it clearer.

**Questions:**

Can the ensemble be used for the proposed method as shown in Table 3?

---

> ### Author Response · Authors · 2023-11-17
> **Author Response to Reviewer PHNg (Part 1/2)**
>
> Thanks for taking the time to review our paper and provide insightful feedback! We address your comments and questions below:
>
>
> **【Confusing Expression】**
>
>
> >*(iii) For “In FSTT-DA, extracting domain-specific knowledge from few-shot data remains a major obstacle. Hence, leveraging generalized semantic knowledge from pre-trained backbone and transferable domain knowledge from source domain datasets is vital.”*
>
> To help the reviewer understand our proposed idea, we provide more contexts and explain our proposed approach in a more intuitive way with concrete examples. On a conceptual level, our proposed approach aims to extract domain-specific knowledge from an unseen target domain using only a few-shot unlabeled data. Specifically, there are three types of knowledge in our proposed framework.
>
> **Type 1**: The generalized knowledge stored in the frozen pre-trained foundation model CLIP that can be generalized to any downstream dataset (e.g. PACS, DomainNet, IwildCam, FMoW, etc).
>
> **Type 2**: The multi-source domain knowledge of a particular downstream dataset that can be learned during training, such as five source domains in Domainnet.
>
> **Type 3**: The target domain-specific knowledge that is unseen during training, such as the leave-one target domain in Domainet.
>
> As mentioned in P2 in the introduction, because of the challenges to extracting target domain-specific knowledge from only few-shot unlabeled data (Type 3), we need to rely on the help of the knowledge that can generalize across all scenarios (Type 1) and the knowledge that can be transferred from multi-source domains (Type 2).
>
>
> >*(ii) they first mention “propose an approach on top of the pre-computed features of the foundation model.”, but then say, “We build a knowledge bank to learn the transferable knowledge from source domains.”. It cannot see the connection between these two sentences.”*
>
> To leverage the out-of-distribution generalized knowledge (Type 1), we first use the pre-computed generalized features of the foundation model without the finetuning on any downstream dataset. In other words, pre-computed features are irrelevant to any source domain. Additionally, to further leverage the multi-source domain knowledge (Type 2) from a particular downstream dataset, we build our approach on the generalized visual features. Specifically,  we propose a knowledge bank to store that multi-source domain knowledge and a mechanism to learn how to transfer that knowledge to a specific target domain via meta-learning.
>
>
> >*(i) I am not sure how the method works from “The domain prompt then directs the visual features towards a particular domain via guidance module.”*
>
> As mentioned in Visual Prompt in related work, a recent trend in prompting has demonstrated that a large foundation model can be guided to adapt to a downstream task by only inserting prompt vectors that encode task-specific knowledge into the feature or input space, without finetuning the entire model. Inspired by those works, we claim that the domain-specific knowledge is stored in the domain prompt. In order to guide the foundation model to adapt to a specific domain without finetuning, the domain prompt is directly injected into the generalized visual features of that foundation model. The description of this guidance mechanism in detail can be found in the “domain guidance module” in section 4.2
>
>
> **【The Selection of CLIP Image Encoder】**
>
> >*The paper uses two different image encoders for DomainNet and WILDS. Could the authors explain the reason? It would be nice to see the performance of two encoders on two datasets.*
>
> To preserve privacy and reduce memory overhead, our proposed method is built on the foundation model without requiring knowledge of its internal architecture or access to its parameters. Consequently, unlike previous works such as FLYP and CLIPood that rely on fine-tuning, the efficacy of our approach heavily depends on the out-of-distribution (OOD) generalization capabilities of the foundation model (please refer to limitations in Appendix A). To handle more challenging benchmark datasets including greater diversity and imbalances in both categories and domains, such as WILDS (see Appendix C), scaling up the model size is a logical step to enhance the foundation model's capacity. Therefore, we choose CLIP-B-16 for the less complex benchmark dataset Domainnet, and CLIP-L-14 for the more challenging WILDS.

---

> ### Author Response · Authors · 2023-11-17
> **Author Response to Reviewer PHNg (Part 2/2)**
>
> **【Missed Experiments】**
>
> >*(i) One SOTA work is not compared, e.g., “CLIPood: Generalizing CLIP to Out-of-Distributions, ICML 2023”.*
>
> Thank you for highlighting the state-of-the-art work of CLIPooD. While CLIPooD provides valuable insights, it is different from our proposed approach and not suitable for direct comparison. First, CLIPooD employs a robust finetuning-based method that involves updating the CLIP model. In contrast, our approach is designed to work without modifying the CLIP model. This key difference is critical, particularly for on-device deployment scenarios, where updating foundational models like CLIP can be impractical due to computational and storage constraints. Moreover, CLIPood employs a Vision-Language architecture. As mentioned in the 'Foundation Models for Domain Generalization' section of the related work, previous works relying on text prompts were constrained by the Vision-Language architecture, leading to numerous limitations. We describe those drawbacks in more detail in the following.
>
>
> 1. CLIPood is restricted to classification tasks and fails in regression problems because it depends on the text encoder from CLIP to initialize the weights of the classifier head. In contrast, our proposed method, which starts from a randomly initialized head, is suitable for regression problems, such as 'PovertyMap' shown in Table 1.
>
> 2. To generate the weights for the classifier head, CLIPood relies on a combination of a template and category names in the text. However, not all classes can be effectively described through natural language. Some classes are best represented by numerical labels. For instance, in a medical lab, researchers classifying images of newly discovered cells or viruses might rely on numerical labels (e.g., 0, 1, 2, ... N) for these as-yet-unnamed categories.
>
> 3. While CLIPood is a robust fine-tuning method specific to the CLIP model, it lacks versatility with other vision foundation models, such as DINOv2. Our approach, however, is more generalized and has been successfully evaluated on DINOv2. Due to the limitations of Vision-Language architectures in previous studies, we were only able to compare our results with two baselines: linear probing and ERM. As illustrated in the subsequent table, our method significantly outperforms these baselines.
>
> |    DINOv2 - ViT-B14       | Clip       | Info       | Paint      | Quick      | Real       | Sketch     | Avg. |
> | ------------------- | ---------- | ---------- | ---------- | ---------- | ---------- | ---------- | ---- |
> | Fintuning (ERM)      | 75.4       | 35.7       | 63.2       | 15.8       | 77.6       | 62       | 55.0 |
> | Linear probing                 | 74.4 | 37.1 | 66.5 | 13.4 | 78.1 | 65.9 | 55.9 |
> | VDPG (Ours)                | 77.2 | 39.6 | 68.1 | 17.6 | 80.4 | 67.7 | 58.4 |
>
>
>
>
>
> >*(ii) It would be nice to see the computational cost of MIRO.*
>
> Please find the computational cost for MIRO in the table below:
>
> |          | Model size | Finetune | Inference (512 images) |
> | -------- | ---------- | -------- | ---------------------- |
> | FYLP     | 149M       | ✓        | 26.2T FLOPS            |
> | DoPrompt | 88.52M     | ✓        | 36.7T FLOPS            |
> | **MIRO**     | 172.57M    |    ✓      | 35.99 T FLOPS          |
> | VDPG     | 102M       | ✗        | 19.7T FLOPS            |
>
> **【Some Suggestions】**
>
> >*The subfigures in Figure 2 do not show the obvious differences between the baseline and the proposed method. It would be nice to change the style or type of figures to make it clearer.*
>
>
> Thank you the the suggestion, we have changed the background to white for better contrast. We also increase the scatter size. We also manually circled some clusters to show that the guided features are more clustered compared to the original CLIP feature.
>
> **【Question】**
>
> >*Can the ensemble be used for the proposed method as shown in Table 3?*
>
> We conduct such an ensemble method by randomly selecting 3 models with different performances:
>
> |  iWildCam metric   | ID Acc  | ID F1  | OOD Acc  | OOD F1  |
> |-----|------------|------------|------------|------------|
> | Model #1  | 77.4 | 60.3 | 79.0 | 46.8 |
> | Model #2  | 76.8 | 59.5 | 78.5 | 46.6 |
> | Model #3  | 75.7 | 54.6 | 77.8 | 44.4 |
> | Ensembled | 76.7 | 59.7 | 78.6 | 46.9 |
>
> As reported in the above table, ensembling has less effect compared with FLYP in Table 3. Please note, that FLYP finetunes the whole image encoder in the source domain and linearly combined with the original CLIP image encoder in the parameter space. The intuition is that the original CLIP has strong OOD capability and the finetuned version contains the data-specific knowledge learned from source domains. The ensemble can be viewed as a fusion of those two sets of knowledge,
>
> However, in our case, we do not finetune the CLIP model, meaning our main source of semantic information from CLIP is the same across models. Therefore, such an ensemble has less effect in our case.

---

> ### Author Response · Authors · 2023-11-22
> **Follow up with Reviewer PHNg**
>
> Dear Reviewer PHNg,
>
> Since today is the last day for author/review discussion, we would like to follow up on our response regarding the questions/concerns you have raised about our paper. If you have further questions or queries, we are ready to answer them promptly today. If you find our response has addressed your concerns, we would be thankful if you could consider sending a confirmation and revise your ratings accordingly if appropriate.
>
> Best,
>
> Authors

---

### Official Review · Reviewer_1gy7 · 2023-10-30

**Soundness:** 3 good
**Presentation:** 3 good
**Contribution:** 3 good
**Rating:** 6
**Confidence:** 4

**Summary:**

This paper solves test-time model adaptation. The authors propose to build a knowledge bank from source domains, and use a domain prompt generator to get a domain-specific prompt conditioned on few-shot target data. A guidance module with domain prompt, a domain-aware constrastive loss and meta-learning are designed to facilitate effective model adaptation. Experiments are conducted on 5 large-scale benchmarks, showing improved performances.

**Strengths:**

- The studied task, Few-Shot Test-Time Domain Adaptation, is important yet challenging. Although there are many works focusing on parameter efficient learning with few-shot data, how to extract useful domain knowledge is an interesting perspect that deserves more research.
-  The authors design several modules including transferable knowledge bank, conditional domain prompt generator and domain-aware contrastive loss and domain guidance module. Overall, the motivation of each module is clear. In addition, a meta-learning based training scheme is proposed for domain episodic training.
- For the experiments, improvements of VPG over comparison methods look siginificant from Tables 1-2. Abalation studies are extensive.
- The writing of paper and figure illustration are good.

**Weaknesses:**

- The method is a bit complex. The loss term in Eq. (5) consists of three terms that need to be properly balanced. As those correlation loss and contrastive loss are at different scales, the model sensitivity against such hyper-parameters especially considering the few-shot data is unclear.
- Since the VDPG is built upon prompt learning of CLIP, comparison with CNNs backbones is less informative. Methods like ERM, CORAL, MTL can be applied with CLIP.
- Fig. 2(a,b) uses a black background, which leads to a poor visual contrast.

**Questions:**

- Since the method learns a conditional domain prompt generator from source domains, I wonder how different numbers of source domains affect the learned generator. In particular, if there is only one source domain, would the method still work?
- In the Bi-direction cross-attention module, what is the benefit using two 'Image attend to tokens' layers?
- In the Algorithm1, how to judge the stopping criterion of convergence?
- In Conditional domain prompt generator E(x), what is 'l is the number of embeddings'? Why not using the image embedding after average pooling for each image to calculate K,V in Attention?

---

> ### Author Response · Authors · 2023-11-17
> **Author Response to Reviewer 1gy7 (Part 1/2)**
>
> We would like to thank you for the valuable comments to make our paper stronger. We address the concerns below:
>
> > *Weakness #1: The method is a bit complex. The loss term in Eq. (5) consists of three terms that need to be properly balanced. As those correlation loss and contrastive loss are at different scales, the model sensitivity against such hyper-parameters especially considering the few-shot data is unclear.*
>
> Incorporating multiple loss functions has been common in training deep models, such as multi-task learning, adding regularization terms, etc. To evaluate the sensitivity on the balancing weights $\lambda$ and $\gamma$, we run experiments with different values while keeping the other one fixed as 0.1:
>
> | $\lambda$        | 0.1        | 0.5        | 5          |
> |------------------|------------|------------|------------|
> | iWildCam OOD F1  | 46.5       | 46.3       | 43.9       |
>
> | $\gamma$         | 0.1        | 0.5        | 5          |
> |------------------|------------|------------|------------|
> | iWildCam OOD F1  | 46.5       | 46.1       | 44.1       |
>
> As reported, our method is not very sensitive about the balancing weights, unless $\lambda$ and $\gamma$ are too large. We would like to mention that, instead of applying all the losses to the final output, our loss terms are applied at different locations of the whole pipeline, each constraining different modules. For example, the correlation loss will not affect the generator and guidance module. However, the latter two will affect the former one. Setting the balancing weights within a small range will not disturb the training, as they will be converged at different training stages.
>
> > *Weakness #2: Since the VDPG is built upon prompt learning of CLIP, comparison with CNN backbones is less informative. Methods like ERM, CORAL, and MTL can be applied with CLIP.*
>
> Given that CORAL (2016) is a relatively newer study and outperforms MTL (2011) when using CNN backbones, we replace the CNN with CLIP-B16 in CORAL for our experiment on DomainNet. Specifically, we adapted the Domainbed codebase with OpenAI's official CLIP implementation, following all the default parameters. As demonstrated in the table below, CORAL achieves better performance than ERM, yet it falls short of more recent works such as MIRO (2022) and our own (2023).
>
> |             | Clip       | Info       | Paint      | Quick      | Real       | Sketch     | Avg. |
> | ----------- | ---------- | ---------- | ---------- | ---------- | ---------- | ---------- | ---- |
> | ERM         | 68.0 (0.1) | 22.5 (0.6) | 46.5 (4.2) | 18.5 (0.9) | 58.7 (2.7) | 52.5 (1.2) | 44.4 |
> | MIRO (2022)        | 74.9 (0.2) | 37.1 (0.4) | 59.8 (0.6) | 18.7 (1.2) | 72.2 (0.2) | 61.2 (0.9) | 54.0 |
> | CORAL (2016)      | 72.1(0.3)  | 29.3(0.4)  | 53.5(0.0)  | 20.7(1.1)  | 65.8 (0.1) | 57.6 (0.0) | 49.8 |
> | VDPG (Ours) | 76.3 (0.2) | 49.3 (0.1) | 67.8 (0.1) | 17.4 (0.2) | 81.5 (0.3) | 66.6 (0.2) | 59.8 |
>
> > *weakness #3: Fig. 2(a,b) uses a black background, which leads to poor visual contrast.*
>
> Thank you for the suggestion. We have changed Fig. 2(a,b) to a white background and increased the size of the scatters for better visualization. We also manually circled some clusters for better comparison. The classes belong to pink, red, and yellow have multiple distinct clusters. One of the red clusters (top one) also separates the brown class into two clusters. But in the guided feature, those classes are more concentrated and can be fitted within one circle.

---

> ### Author Response · Authors · 2023-11-17
> **Author Response to Reviewer 1gy7 (Part 2/2)**
>
> > *Question #1: Since the method learns a conditional domain prompt generator from source domains, I wonder how different numbers of source domains affect the learned generator. In particular, if there is only one source domain, would the method still work?*
>
> To show the effect from different numbers of source domains, we conduct such an experiment by treating Clipart as the target domain and randomly selecting the source domains:
>
> | Number of source domains | 1     | 2     | 3     | 4     | 5     |
> |--------------------------|-------|-------|-------|-------|-------|
> | Accuracy on Clipart      | 55.8  | 71.8  |74.5   |75.7   |76.3   |
>
> When there is only one source domain, we are not able to compute domain contrastive loss, therefore, the capability to generate domain-specific knowledge is hampered. When there are two domains, the generator is enforced to compute domain-specific knowledge, therefore the accuracy is boosted. When more and more source domains are involved during training, the generalization is improved but the overall gain becomes smaller.
>
> > *Question #2: In the Bi-direction cross-attention module, what is the benefit t using two 'Image attend to tokens' layers?*
>
> A stacked bidirectional conditional transformer layer has shown effectiveness in other tasks for better information fusion, such as segmentation (SAM (Kirillov et al., 2023)). By "image attend to tokens", we mean that we treat the prompt token (domain-specific information) as a condition and query the relevant information from image embeddings. By applying the second "image attend to tokens", we can ensure that the final output is always conditioned on the extracted domain-specific information. We also ran an experiment by replacing the guidance module with regular transformer layers and performing the pre-pending operation without conditions:
>
> |                 | F1 score on iWildCam|
> |-----------------|---------------------|
> | Bidirectional condition layers| 46.4  |
> | Regular self-attention layers | 39.7  |
>
> As reported from the above table, conditional operation plays a paramount role in the domain guiding process.
>
> > *Question #3: In the Algorithm1, how to judge the stopping criterion of convergence?*
>
> We use early stop when the validation performance is not improved by 3 epochs. Since we use a cosine learning rate (LR) scheduler, the LR is quite small after 20 epochs. Most of the experiments are finished within 30 epochs.
>
> > *Question #4: In Conditional domain prompt generator E(x), what is 'l is the number of embeddings'? Why not use the image embedding after average pooling for each image to calculate K, V in Attention?*
>
> In a vision transformer, the images are partitioned into small patches, and each patch is transformed into an embedding. "I" represents the number of patches. Average pooling in the "I" dimension means the average of all the patch embeddings which may diminish some information. Keeping the "I" dimension is a common operation in the cross-attention mechanism as in Perceiver (Jaegle et al., 2021).

---

> > ### Comment · Reviewer_1gy7 · 2023-11-22
> > **response to rebuttal**
> >
> > I appreciate the authors' response to my questions. I keep my original score as positive.

---

> ### Author Response · Authors · 2023-11-22
> **Follow up with Reviewer 1gy7**
>
> Dear Reviewer 1gy7,
>
> Since today is the last day for author/review discussion, we would like to follow up on our response regarding the questions/concerns you have raised about our paper. If you have further questions or queries, we are ready to answer them promptly today. If you find our response has addressed your concerns, we would be thankful if you could consider sending a confirmation and revise your ratings accordingly if appropriate.
>
> Best,
>
> Authors

---

### Official Review · Reviewer_f7Jt · 2023-11-01

**Soundness:** 3 good
**Presentation:** 3 good
**Contribution:** 3 good
**Rating:** 8
**Confidence:** 4

**Summary:**

In this work, authors leverage the foundation model CLIP to provide test time adaptation on a new target domain, while only relying on a few unlabeled samples at test. Different components are used to attain state-of-the-art performance on the standard benchmarks. Transferable domain knowledge bank and domain-specific prompt generation to guide the visual features from the new domain to perform classification in the known label space. This work has extensive experiments and ablations to show the efficacy of their method.

**Strengths:**

1. Well-written and easy to follow
2. Interesting approach to leverage CLIP knowledge for performing few-shot TTA. The different novel components are combined together in a non-trivial manner to attain the SoTA performance.
3. Extensive ablations on knowledge bank, prompt generation, losses and evaluation on the standard benchmarks are provided.

**Weaknesses:**

1. How low can KB size be? or how much sensitivity to Z value we can have?

**Questions:**

I have my minor comment in the weakness section.

---

> ### Author Response · Authors · 2023-11-17
> **Author Response to Reviewer f7Jt**
>
> Thank you for your constructive comments, we address your concerns below:
>
> > *How low can KB size be? or how much sensitivity to Z value we can have?*
>
> To show the effect of the size of the knowledge bank (KB), we conduct experiments on iWildCam with various KB sizes as shown below:
>
> | KB size | 1     | 50     | 100     | 150     | 200     |
> |--------------------------|-------|-------|-------|-------|-------|
> | iWildCam OOD F1 | 41.6  | 44.0  |46.5   |44.9   |44.3   |
>
> As reported, a larger KB size can better learn the transferable knowledge from the source domain. But setting KB too large cannot guarantee a better performance.
>
> Our method freezes the CLIP model to preserve its strong OOD generalization capability. Therefore, the knowledge bank is supposed to learn the transferable knowledge of the source domain, which represents the downstream dataset. As a result, the size of KB is highly relevant to the source domain data. On the other hand, our design is domain-centric, learning to adapt to each target domain. Ideally, each vector in the KB is expected to learn knowledge corresponding to one source domain. However, as some source domains are correlated, their domain-specific knowledge can be shrunk. As shown above, there are 243 source domains in iWildCam, but the performance is more optimal when the KB size is close to 100. Table 11 in the Appendix shows the KB size for other benchmarks.

---

> ### Author Response · Authors · 2023-11-22
> **Follow up with Reviewer f7Jt**
>
> Dear Reviewer f7Jt,
>
> Since today is the last day for author/review discussion, we would like to follow up on our response regarding the questions/concerns you have raised about our paper. If you have further questions or queries, we are ready to answer them promptly today. If you find our response has addressed your concerns, we would be thankful if you could consider sending a confirmation and revise your ratings accordingly if appropriate.
>
> Best,
>
> Authors

---

### Official Review · Reviewer_hhmW · 2023-11-02

**Soundness:** 3 good
**Presentation:** 3 good
**Contribution:** 3 good
**Rating:** 5
**Confidence:** 2

**Summary:**

The paper introduces a new approach to address distribution shifts at test-time by utilizing a few unlabeled data, emphasizing the challenge of extracting domain knowledge from limited data. The proposed method builds on the features of foundation models, creating a knowledge bank to learn transferable knowledge from source domains. When given few-shot target data, a domain prompt generator is developed to condense this knowledge bank into a domain-specific prompt, which guides the visual features to a specific domain. The method, named Visual Domain Prompt Generator (VDPG), integrates a domain-aware contrastive loss and employs meta-learning to improve domain knowledge extraction, and it outperforms previous methods on multiple benchmarks.

**Strengths:**

- VDPG not only outperforms other methods on the WILDS benchmark, but it specifically showcases superior results on individual datasets like iWildCam, Camelyon17, and FMoW.
- VDPG's ability to generate high-quality domain-specific prompts tailored to each target domain is not just a novel approach, but one that proves effective
- When compared to methods like FYLP and DoPrompt, VDPG is designed for quicker adaptation and inference, making it a more feasible choice for real-world applications where computational resources and time are essential factors.

**Weaknesses:**

- The results show that ERM training alone cannot drive performance. Specific configurations, such as episodic learning, are necessary to boost performance. This indicates a complexity in training dynamics that might be challenging to replicate or optimize in varied settings.
- The method adapts with few-shot data before making inferences on all target data. While this is computationally efficient, there's a potential risk of overfitting or being overly reliant on a limited subset of data.

**Questions:**

.

---

> ### Author Response · Authors · 2023-11-17
> **Author Response to Reviewer hhmW (Part 1/2)**
>
> > *Q2: The method adapts with few-shot data before making inferences on all target data. While this is computationally efficient, there's a potential risk of overfitting or being overly reliant on a limited subset of data.*
>
> Thank you for highlighting the critical challenge of overfitting in Few-shot Test-time Adaptation. This challenge has been a key driver behind the motivation for both previous works [1, 2] and our own, leading us to adopt a meta-learning framework. Meta-learning is a widely recognized and effective approach for addressing overfitting in few-shot image classification tasks, as detailed in [3].
>
> Given the scarcity of domain-specific information within the few-shot target domain, our approach needs to rely on the assistance of the shared knowledge transferred from multiple source domains. To facilitate this knowledge transfer, we have introduced a knowledge bank, where sharable knowledge is stored. Additionally, we have proposed a domain prompt generator that plays a crucial role in determining which subset of the shared knowledge can be transferred to the target domain based on the limited few-shot target data. This process ultimately results in the generation of a domain-specific prompt tailored to the target domain.
>
> To ensure that both the shared knowledge stored in the knowledge bank and the mechanisms of knowledge transfer can be generalized effectively across diverse domains, we have leveraged episodic learning, namely meta-learning. This approach allows us to sample a wide range of domains, each with varying combinations of data examples.
>
> While Section 4.2 and Algorithm 1 provide a detailed explanation of episodic learning, we aim to offer additional context here to help reviewers grasp the intuition of a meta-learning framework for mitigating overfitting.
>
> In standard supervised training (ERM), we typically sample a mini-batch of data examples during each training iteration, with the expectation that the model will learn to generalize across a diverse range of data examples. However, episodic learning introduces a different perspective. During each training iteration, we begin by sampling a mini-batch of domains. From each of these domains, we further sample two disjoint sets of data examples: the first set is unlabeled in few-shot, known as the support set, while the second set is labeled in many-shot, known as the query set. Within each domain, we employ the unlabeled support set as input for a domain generator, which generates a domain-specific prompt. This prompt is then evaluated using the labeled query set. The underlying rationale for this step is to encourage the domain prompt generator to extract an effective domain-specific prompt based on the few-shot unlabeled support set, with the expectation that it will perform well on unseen query sets from the same domain.
>
> Through numerous iterations of episodic learning, the model gradually grasps the mechanism across various domains, learning how to generate a domain-specific prompt even when given access to only a limited amount of unlabeled data. That means our prompt generation process is designed to be both class and data instance agnostic, which helps to avoid the pitfalls of overfitting. The effectiveness of this design is validated in Fig. 2 (c-d) and Fig. 4 of the Appendix.
>
> [1] Adaptive risk minimization: Learning to adapt to domain shift. NeurIPS 2021
>
> [2] Meta-dmoe: Adapting to domain shift by meta-distillation from mixture-of-experts. NeurIPS 2022
>
> [3] Meta-learning in neural networks: A survey. PAMI 2021

---

> ### Author Response · Authors · 2023-11-17
> **Author Response to Reviewer hhmW (Part 2/2)**
>
> > *Q1: ... ERM training alone cannot drive performance ... episodic learning is necessary to boost performance. ... but challenging to replicate or optimize in varied settings.*
>
> Thank you for pointing out the complexity of episodic learning compared to ERM. However, it's important to clarify that the implementation and the additional hyperparameters involved are not overly complex. As mentioned in the previous response, the primary distinction between those two learning paradigms lies in the data sampling process.
>
> In ERM, we simply sample a mini-batch of data instances during each iteration. In contrast, episodic learning employs a hierarchical sampling approach by first sampling a mini-batch of domains and subsequently sampling the support and query sets from each of these domains. The model is updated based on query loss, with detailed descriptions of the loss functions provided in Section 4.1.
>
> It's worth noting that there may be a potential misunderstanding regarding meta-learning, possibly conflicting it with MAML [4], which requires the computation of second-order gradients during optimization. Meta-learning is a broader framework encompassing various categories of algorithms, and MAML is just one specific optimization-based meta-learning. In our proposed approach, we do not employ optimization-based techniques and, therefore, do not encounter complex optimization issues.
>
> Furthermore, to facilitate easier replication of our proposed method by other researchers, we plan to release our code as open source upon paper acceptance.
>
> [4] Model-Agnostic Meta-Learning for Fast Adaptation of Deep Networks. ICML 2017

---

> ### Author Response · Authors · 2023-11-22
> **Follow up with Reviewer hhmW**
>
> Dear Reviewer hhmW,
>
> Since today is the last day for author/review discussion, we would like to follow up on our response regarding the questions/concerns you have raised about our paper. If you have further questions or queries, we are ready to answer them promptly today. If you find our response has addressed your concerns, we would be thankful if you could consider sending a confirmation and revise your ratings accordingly if appropriate.
>
> Best,
>
> Authors

---

### Author Response · Authors · 2023-11-17
**Thanks to all the reviewers!**

We thank reviewers for their positive feedback on our paper which contributes to:

* VDPG, a novel visual prompting approach in Few-Shot Test-Time Domain Adaptation, which formulates the adaptation process as generating a visual prompt that encodes domain-specific knowledge to guide the foundational model

* state-of-the-art results on the standard large-scale distribution shift benchmark including DomainNet and WILDS

All the reviewers agreed that our work provides an effective solution to alleviate the challenges of Few-Shot Test-Time Domain Adaptation. They praised the novelty of our approach [hhmW, f7Jt, 1gy7], its clarity [f7Jt, 1gy7], the comprehensiveness of the experiments and ablation study [hhmW, f7Jt, 1gy7, PHNg], as well as the additional advantages it offers in terms of efficiency for resource-constrained real-world applications [hhmW].

Primarily, the reviewers requested additional experiments on hyper-parameters and sought further explanation on specific sub-components of our proposed approach, asking us to provide more in-depth details. Consequently, in our rebuttal, we have provided additional context and explained our idea in a more intuitive manner [hhmW, PHNg]. Furthermore, we have elaborated on the experimental settings and conducted new experiments [f7Jt, 1gy7, PHNg].

We believe that these experiments and our responses successfully address all the questions, concerns, and comments raised by the reviewers.

---

### Meta-Review · Area_Chair_S4iu · 2023-12-06

**Metareview:**

The paper receives mixed ratings: 1 accept, 1 borderline accept, and 2 borderline reject. Originally the main concerns from the reviewers are: 1) some technical clarifications; 2) complexity of the proposed framework; 3) more experiments (e.g., hyperparameter sensitivity, different backbone, number of source domains). The AC took a close look at the paper and finds that most critical issues are addressed well in the rebuttal, especially for reviewer hhmW and PHNg about more experiments and technical clarity. Therefore, based on the current state, the AC recommends an acceptance rating, in which the authors should incorporate additional materials in the final version with suggested feedback from the reviewers.

**Justification For Why Not Higher Score:**

While the idea and the proposed method is of great interest to the few-shot test-time domain adaptation problem, the authors still need to include many materials suggested by the reviewers to make it stronger.

**Justification For Why Not Lower Score:**

N/A

---

### Decision · Program_Chairs · 2024-01-16

Accept (poster)